# Role of SHH in Patterning Human Pluripotent Cells towards Ventral Forebrain Fates

**DOI:** 10.3390/cells10040914

**Published:** 2021-04-16

**Authors:** Melanie V. Brady, Flora M. Vaccarino

**Affiliations:** 1Child Study Center, Yale University, New Haven, CT 06520, USA; melanie.brady@yale.edu; 2Department of Neuroscience, Yale University, New Haven, CT 06520, USA; 3Yale Kavli Institute for Neuroscience, New Haven, CT 06520, USA

**Keywords:** iPSC, organoids, basal ganglia, interneurons, ventral telencephalon, sonic hedgehog

## Abstract

The complexities of human neurodevelopment have historically been challenging to decipher but continue to be of great interest in the contexts of healthy neurobiology and disease. The classic animal models and monolayer in vitro systems have limited the types of questions scientists can strive to answer in addition to the technical ability to answer them. However, the tridimensional human stem cell-derived organoid system provides the unique opportunity to model human development and mimic the diverse cellular composition of human organs. This strategy is adaptable and malleable, and these neural organoids possess the morphogenic sensitivity to be patterned in various ways to generate the different regions of the human brain. Furthermore, recapitulating human development provides a platform for disease modeling. One master regulator of human neurodevelopment in many regions of the human brain is sonic hedgehog (SHH), whose expression gradient and pathway activation are responsible for conferring ventral identity and shaping cellular phenotypes throughout the neural axis. This review first discusses the benefits, challenges, and limitations of using organoids for studying human neurodevelopment and disease, comparing advantages and disadvantages with other in vivo and in vitro model systems. Next, we explore the range of control that SHH exhibits on human neurodevelopment, and the application of SHH to various stem cell methodologies, including organoids, to expand our understanding of human development and disease. We outline how this strategy will eventually bring us much closer to uncovering the intricacies of human neurodevelopment and biology.

## 1. Modeling of CNS Development and Disease In Vivo and In Vitro

The development of the human central nervous system (CNS) is a complex series of sequential molecular events that guide cellular proliferation, differentiation, and migratory patterns to build the organization of what are the most advanced neural circuits in the animal kingdom. These processes generate a vast cellular diversity integrated into networks that ultimately produce higher-order human functions, from cognition and motor coordination to consciousness.

The inherent complexity of the human CNS has historically complicated scientists’ ability to model and understand all of its parts, thereby making the study of developmental and degenerative abnormalities equally challenging. While animal models have long served as tools to uncover human neurobiology, their limitations have continued to emphasize the need for a reliable human model system where species-specific developmental trajectories and healthy and disease pathophysiology can be convincingly captured. For instance, the basal ganglia plays an essential role in both learning [1,2] and motor processes [3], and it has been implicated in a broad range of neuropsychiatric disorders; because of this, this subpallial region and its circuitry have been investigated in many animal models, from rodents to primates. While general cortico-basal ganglia regional architecture is maintained across rodents and humans, the globus pallidus pars interna, termed the entopeduncular nucleus in rodents, and the frontal cortex are both comparatively immature structures in rodents [4]. Basal ganglia output also varies between humans and rodents, with extensive output to the thalamus versus midbrain and brainstem regions, respectively [5]. Furthermore, certain cell types, such as the outer radial glia of the dorsal telencephalon, that are essential to human cortical development are substantially lacking in rodents [6]. Additionally, disease modeling in animals often requires genetic engineering to mimic human dysfunction that does not naturally occur, and leads to animal behaviors that are sometimes difficult to characterize in the context of human phenotypes. Higher order processing achieved by the human brain also suggests the possibility for equally complex and intricate abnormalities, and represents yet another reason why a human-specific model system has become an empirical necessity.

Human induced pluripotent stem cell (iPSC)-derived organoids are tridimensional multilayered aggregates with the intrinsic capability to capture human embryonic development. For this reason, organoids present a tremendous opportunity to model both human development and disease to further not only our biological understanding but also clinically relevant investigations, from diagnostics to therapeutics. Human iPSC technology came during a time of great ethical distress over the use of embryonic stem cells, which gained empirical traction after the use of murine pluripotent cell lines [7,8]. Subject-specific iPSCs could be generated from the initial harvest of a simple adult human fibroblast. The exposure of fibroblasts to a cocktail of reprogramming factors reverts their development back to pluripotency, from which point the cells can be differentiated into any cell of the human body [9,10].

There are two systems generally used to generate neuronal cells from iPSCs: monolayers and organoids. Neural iPSC differentiation protocols begin first with the single layered iPSCs. The cells are placed in a physical environment that fosters their assembly into a tridimensional aggregate. The aggregates become embryoid bodies—named for their ability to become any of the three germ layers, the endoderm, mesoderm or ectoderm. Guided by a set of early morphogens and with further culturing, these aggregates can be pushed to adopt a neural fate and express a diverse array of progenitors mimicking early regional patterning of brain tissue [11]. Once the regional identities are specified, withdrawal of mitotic factors from the media fosters post-mitotic differentiation of various cell types that reside within said regions. These aggregates, which have been called “three-dimensional aggregation culture (SFEBq)” [11,12] and more recently organoids [13], are self-organizing and their morphological growth occurs spontaneously. Studies have found that neural aggregates inherently adopt a cortical (dorsoanterior) telencephalic fate by default [12,13,14,15], similar to the developing cerebral cortex. We can capture a broader range of human anatomy, however, by introducing critical factors that endogenously pattern the CNS into various regions. Exposure to early morphogens and subsequent cellular differentiation can also be accomplished in iPSC grown in dense 2D monolayers [16]. For detailed differences between monolayers and organoid culture protocol, see recent reviews [17,18,19].

The monolayer system has certain advantages. Cells cultured in a 2D monolayer system are more easily exposed compared to those within the 3D organoid network, which provides greater cellular access for electrophysiological experiments. Additionally, their plating allows for the observation of neurite outgrowth, and their easier expansion yields the generation of cell types in high concentrations, providing sufficient material and also reliability when analyzing cellular morphology. Furthermore, cells grown in the 2D monolayer system are quicker to culture and are postulated to be more reproducible. On the other hand, there are many limitations of the 2D system still to consider. The over-arching theme is that the 2D system does not closely recapitulate the physiological complexity of in vivo human developmental biology. The 2D system favors the development of more basic neuronal cellular populations generally lacking in vivo cell type diversity and complexity, mostly due to the absence of microenvironmental factors, including chemical morphogens and physical/mechanical interactions among cells. The variety of functions that a single cell type possesses is a combined result of its gene expression, the cellular network it resides within and the other cell types with which it communicates. In a recent study directly comparing 2D (monolayer) and 3D (organoid) systems, we found that laminin-mediated integrin signaling was upregulated in monolayers as compared to organoids, while the reverse was true for Notch signaling and cell adhesion molecules such as cadherins. In vivo, laminin is a major extracellular matrix (ECM) component of basement membranes and septa that separate major tissue compartments, but is generally not present when cells directly contact each other via their outer membrane. Interesting, laminin is commonly used as coating to favor cell adhesion to the plastic dish in 2D cultures. Downregulating laminin-stimulated integrin signaling reversed excessive cell proliferation of monolayer cultures [20]. Thus, excessive integrin signaling results in biological deficiencies which compromise their ability to differentiate [20]. These differentiation impairments can also arise from an impairment in direct cell-to-cell interactions and Notch signaling among precursors and daughter cells [20].

Another defining feature of the organoid system that elevates its potential beyond animal models is its subject-specific design. From a simple non-invasive skin biopsy [10], or more recently urine [21,22] or blood samples [23,24,25], the patient’s cells are reprogrammed into iPSCs and cultured to become neural organoids while maintaining the subject’s specific genetic background. This individualized nature carries the potential to reveal brain cytoarchitecture that resembles that of the person from whom the cells originally came. This opens up another possibility, especially when considering disease modeling and medical treatment plans, paving a way towards precision medicine. Beyond disease modeling, organoids have even been utilized to compare developmental differences between modern humans and the extinct Neanderthals to identify key aspects of human evolution [26]. An additional technical benefit is that, once reprogrammed, the iPSCs can be cultured into perpetuity, essentially creating an immortal system that can generate an unlimited reserve of cells.

At the same time, there are limitations of the organoid system to consider as well, one of which is a certain degree of variability across preparations. Recent attempts to mitigate this concern have focused on the introduction of synthetic scaffolding matrices such as hydrogels to mimic physiological microenvironments suitable for organogenesis [27]. Despite substantial efforts towards protocol optimization to improve organoid reproducibility, variability of cellular composition is nonetheless observed across preparations, raising questions about system robustness. Assessing the extent of variability can also be challenging as organoids require months for optimal growth suitable for harvesting and analyses, making the time-course of organoid maturation a challenge for users of the system. Furthermore, the extent to which number and type of cells expressed within organoids match physiological percentages is still unclear. For additional detailed information on these limitations, see [28,29]. While organoid cultures can be maintained long-term for several months, achieving uniform cellular health in a prolonged lifespan has been equally challenging. Furthermore, some cellular complexities are yet to be implemented in the organoid cultures, including, but not limited to, vascularization of the tissue, myelination of neuronal processes, and organization of cortical layers in pallial structures (for reviews on these limitations, see [30,31,32,33]). Given that the organoids capture embryonic- and fetal-stage development best, deciphering whether organoids are applicable to both neurodevelopmental and neurodegenerative disorders is an ongoing investigation.

## 2. SHH Signaling during Embryogenesis

The basal ganglia arise from the ventral telencephalon, an anterior extension of ventral neuraxis, which is patterned by the sonic hedgehog (SHH) signaling pathway in normal vertebrate development. This dynamic protein serves a multitude of functions in a context-dependent manner, both as a mitogen to promote cellular proliferation, and a morphogen to direct cellular and regional specification towards ventral fates [34,35,36,37]. Embryological investigation has exposed many of the multifaceted roles that SHH plays in development, adding insight into the disorders observed with deficits in SHH expression [35,38,39,40].

Before we focus on the varying ways that SHH has been used to pattern stem cells, the command that this ligand has on instructing neural diversity and cellular organization must first be understood. One of the first SHH-expressing tissues during mammalian embryogenesis is the notochord, which begins its developmental influence in early neural plate stages and crescendos upon neural tube formation [Figure 1] [41,42,43]. SHH emanates from the notochord, and its role as a chief ventralizing signal was originally discovered from transplantation experiments in chick embryos [44,45,46] and in vitro experiments with neural plate tissue explants [43,44,47]; these works were further complemented by genetic experimentation for both gain- [47,48] and loss-of-function [38] SHH signaling. The notochord has immediate contact with the floor plate at the ventral midline of the neural tube, and induces these nearby cells to express SHH [Figure 1B] [49]. The capacity of SHH to program a variety of cellular identities is explicated in a graded fashion, as indicated by in vitro studies (Figure 1B) [50]. The cells of the floor plate secrete SHH and establish a concentration gradient within the CNS with the most elevated protein levels found ventrally; the degree to which different cell types emerge, therefore, is concentration-dependent. There is a unique ‘release and response’ duality of the neural tube that is essential for determining the positioning of cell types along the main coordinates of the neuraxis, whereby a proportion of neural tube cells, found ventrally, express and secrete SHH, while another large portion of neural tube cells respond to the ligand in a concentration-dependent fashion. The SHH concentration gradient ultimately governs the patterning of the central nervous system along the dorsoventral domains. Regulation of SHH expression extends beyond its direct release from the notochord, and includes its repression by bone morphogenic proteins (BMP) and WNT signaling molecules [51,52,53]. SHH has antagonistic interactions with the dorsally expressed BMP and WNT signaling proteins [Figure 1B], which will fully define the limits of the range of each domain. Herein lies the initial guide to the central nervous system’s rise to cellular diversity.

## 3. Cell Types Requiring SHH Signaling in Vertebrate Development and Their Role in Diseases of the CNS

Along the rostrocaudal axis, SHH, in conjunction with other morphogens, induces cell types at in ventral domains at various levels in the forebrain, midbrain, hindbrain and spinal cord (Figure 1A,C). In the forebrain, SHH promotes the differentiation of the ventral telencephalic ganglionic eminences, which contribute to adult basal ganglia formation [55]. In addition, the ventral telencephalic ganglionic eminences are the site of origin of essentially all telencephalic inhibitory interneurons [56]. Therefore, SHH is essential for kick-starting the cascade of signaling that leads to the development of many classes of inhibitory interneurons that eventually populate the forebrain, modulating excitatory neurotransmission and are integral components of an optimally functioning cortico-basal ganglia circuitry that directs motor coordination and cognition. Additionally, SHH governs dopaminergic and serotonergic neuronal fates in the midbrain and hindbrain regions, essential for motor and emotional control, respectively [44,57,58,59]. Ye and colleagues showed that achieving such neuronal diversity is directed by the distinct combination of molecular positional cues, assembled in an informational “grid” system of morphogens [59,60,61]. The developmental grid system of morphogens and fate determinants, first described by Wolpert [60] and further defined by Rubenstein and colleagues [61], organizes regional neural patterning and cell fate essentially creating a topography. Extracellular molecules determine the grid and serve as signaling centers during development. As cells mature, cell type specificity is acquired based on the cells’ precise location within this developmental grid. Ye et al. further investigated this grid system concept in rat ventral midbrain and hindbrain explants [59]. By manipulating exposure to SHH and fibroblast growth factor 8 (FGF8), blocking functionality of the morphogens, and controlling timing of morphogen exposure, Ye and colleagues showed that dopaminergic induction and neuronal commitment throughout the midbrain and forebrain required signaling from both SHH and FGF8. More caudally, introduction of FGF4 signaling pre-patterns the rostral hindbrain, such that subsequent signaling of FGF8 and SHH pushes progenitors to adopt a serotonergic fate instead of dopaminergic [59]. Extending caudally from the hindbrain to the spinal cord, SHH promotes the differentiation of motor neurons and ventral interneurons. Later in development, SHH plays an essential role in oligodendrocyte development at many levels of the neuraxis [62]. These cell types make myelin, which is critical for the speed of neural signaling and integration of sensory information [51,63,64].

Understanding the neurodevelopment of these important cellular lineages provides the necessary foundation from which we can decipher perturbations that yield neurological disorders, of both a neurodevelopmental and neurodegenerative nature. For instance, altered development of dopamine neurons, originating in the substantia nigra of the midbrain and projecting to the basal ganglia of the ventral forebrain, has been linked to both movement and neuropsychiatric disorders, from Tourette syndrome [63,64] to schizophrenia [65,66,67]. Additionally, dysfunction and fluctuations in serotonergic neuron numbers, originating in the hindbrain but whose axonal projections innervate areas throughout the entire brain, are directly implicated in many neuropsychiatric disorders such as anxiety and depression [68,69,70]. In the forebrain, abnormal development of the striatum in the basal ganglia contributes to other movement and psychiatric disorders, from Huntington’s disease to obsessive compulsive disorder (OCD).

## 4. Elucidating the SHH Signaling Pathway Paves the Way for Application in In Vitro Patterning

The fundamental role of SHH and other morphogens, revealed by gain-of-function and loss-of-function studies in mice, has paved the way for scientists to try to identify these complex processes in human models [35,71,72]. Pioneering genetic studies in *Drosophila* identified several key genes that are integral components of SHH signaling [73], most notably *patched, smoothened*, and three separate *gli* genes (*cubitus interruptus* genes in Drosophila). *Patched* (ptc) and *smoothened* (smo) are both transmembrane protein receptors, but mediate SHH signaling in opposing ways, in that ptc inhibits SHH signaling while smo activates it [71,74] (Figure 2). In the absence of SHH, ptc binds smo and constitutively inactivates it, preventing signaling of the pathway. However, in the presence of SHH, the ligand binds ptc, which alters the receptor’s conformation; ptc then becomes physically untethered from smo, relieving its inhibition such that the SHH pathway can become active [72] (Figure 2). Further examination of ptc and smo in vertebrates has suggested their similarly important roles in mediating SHH signaling to pattern the neural tube [71,74].

As will be discussed in the next section, the SHH signaling pathway has been exploited as a vital component of many novel human in vitro strategies to uncover the neurobiology of developmental disorders and to discover novel therapeutic technologies. The remainder of this review will discuss how the community has utilized SHH as a tool to promote neuronal differentiation into specific neuron types and regional development, and the biological and clinical applications of these distinct methodologies. We will briefly discuss the use of SHH in various monolayer stem cell preparations and their limitations before discussing its use and potential in iPSC-derived organoid systems.

## 5. The Use of SHH in In Vitro Disease Modeling

### 5.1. Parkinson’s Disease

To explore a potential cellular repair for Parkinson’s disease [75]. Kriks and colleagues used joint molecular activation of SHH and canonical WNT signaling to enhance specification of human embryonic stem cells (hESCs) towards a dopaminergic neuron fate. They achieved engraftable hESC-derived midbrain dopaminergic neurons that thrived long-term in the striatum of three different Parkinsonian animal model systems including 6-hydroxy-dopamine-ablated mice and rats and 1-methyl-4-phenyl-1,2,3,6-tetrahydropyridine (MPTP)-lesioned rhesus monkeys. Although the behavioral recovery was limited, the survival span of the graft was 1 month, and led to robustly developed fibers positive for tyrosine hydroxylase, an enzyme required for dopamine synthesis.

Another study revealed successful generation of GABAergic medium spiny neurons (MSN) from hESCs exposed solely to SHH treatment, and investigated the potential for cell therapy as a treatment for Huntington’s disease [76]. Ma and colleagues found that treatment of hESCs with a specific concentration of SHH was sufficient to foster lateral ganglionic eminence (LGE)-like development in monolayer neuronal cultures and ultimately MSN development after extended culture. Beyond gene expression, the SHH-treated hESC cultures produced the appropriate striatal MSN morphology, sprouting dendrites with spines, and the differentiated neurons displayed functional properties in the form of spontaneous synaptic activity as well as action potentials subsequent to current injection. Transplantation of these LGE-like progenitors into mice with lesioned striatal tissue confirmed the progenitors’ ability to mature and differentiate into GABAergic DARPP32-positive neurons, with concomitant recovery of volume loss and cellular concentration of the lesioned striatum as early as four months post-transplant.

While Parkinsonian symptoms can be ameliorated by sufficient replenishment of dopamine stores, the complexity of circuitry restoration remains a barrier for developing Huntington’s disease treatments. Long-range projections, characteristic of medium spiny neurons, are required to innervate various target regions of both the basal ganglia and cerebral cortex to regulate the dynamic motor circuits. Both distance and continued cellular access serve as considerable obstacles that the in vitro-generated neurons are yet to overcome. That being said, these findings convey the powerful control SHH has alone to pattern developing neurons to a quasi-mature and functional extent in an animal model.

### 5.2. Motor Neuron Disorders

Another study found that prolonged long-term exposure of iPSCs to SHH could ultimately yield motor neurons able to form neuromuscular connections upon co-culturing with myotubes [77]. Du and colleagues used iPSC-derived SHH-induced motor neurons to model disorders plaguing specifically motor neuron populations, including spinal muscular atrophy (SMA) and amyotrophic lateral sclerosis (ALS). Du et al. capitalized on the patient-specific nature of the iPSC technology and compared patient-derived motor neurons to healthy controls. The expression of the survival motor neuron (SMN) gene was, as expected, decreased in differentiated motor neurons cultured from hiPSCS derived from SMA as opposed to non-SMA control individuals. Furthermore, differentiated motor neurons from ALS individuals carrying the D90A mutation in the superoxide dismutase gene (*SOD1*) revealed a downregulation of *NEFL* gene expression, encoding for the light polypeptide neurofilament protein, consistent with this particular ALS mutation. These findings highlight the ability of SHH to direct the correct generation of cell types equipped with appropriate endogenous machinery, such that this in vitro approach was capable of recapitulating disease phenotypes.

### 5.3. Serotonin Neurons

The potential use of the SHH agonist for discovery and application is further exemplified by yet another recent study utilizing SHH to direct the differentiation of functional serotonin neurons of the raphe nucleus [78]. Using both hESCs and hiPSCs, Lu et al. showed that transient exposure to SHH was sufficient to bias hindbrain progenitors towards a serotonin fate. SHH signaling, along with WNT activation, were the essential modulators used to enrich the cultures towards tryptophan hydroxylase-positive, electrophysiologically functional serotonin neurons.

It is important to note that while both motor neuron [77] and serotonergic neuron [78] generation were obtained under the influence of similar morphogenic cocktails, achieving these distinct neuronal populations was possible by adjusting timing and concentration of these morphogens. For both preparations, it was imperative to first caudalize the cultures with WNT agonists to achieve a posterior fate, before ventralizing the system with SHH. While this is true for both motor and serotonergic neuron generation, motor neurons required 1 µM of the synthetic SHH agonist, purmorphamine, for 6 days [77], while serotonergic neurons required 1000 ng/mL (1.9 µM) for 1 week [78]. The motor neuron preparation also coupled the SHH treatment with retinoic acid to assist in ultimate differentiation, which was absent from the serotonergic preparations. Similar to the endogenous human system, interactions among these differing factors play significant roles in the developmental trajectory and ultimate differentiation of these neuronal populations.

After ascertaining the appropriate gene expression patterns consistent with serotonin neurons [78], the cultures were utilized for pharmacological screening to reveal the potential of the in vitro methodology as a system to test drug candidates for disease treatment. Two classes of serotonin-targeting drugs were tested, including activators of serotonin release and selective serotonin reuptake inhibitors. The treatment of the hiPSC-derived hindbrain serotonin neurons with either class of drug resulted in increased serotonin concentration in both a dose- and time-dependent manner, validating the known pharmacology of the drugs. Therefore, the differentiated serotonin neurons present a potential experimental approach for the examination of drug therapies for a number of neuropsychiatric disorders, including, but not limited to, a variety of eating, sleep and mood disorders.

As discussed in the Lu et al. study, many serotonin-expressing cells in the 2D system lack the cellular mechanics and equipment to function properly. As described in the Du et al. study, motor neuron progenitors cultured in the 2D system have a short lifespan and limited differentiation potential, lasting a mere two to five passages before losing their potency. These limitations make the use of the cells for disease modeling, drug screening and cellular replacement unpredictable.

Furthermore, the aforementioned studies have had few or no follow-up investigations, which raises the question of additional complications in reproducibility, reliability and applicability of these systems. It is important to note that serotonergic neuronal generation was followed up by additional characterization of the serotonergic neurons post-transplantation in the mouse brain [79]. While the transplanted neurons were found to project to various regions of the hindbrain, their projections were limited and were not seen to reach the anterior regions of the brain. Thus, although this transplantation survival shows promise, future follow-up studies are necessary before the 2D model systems can be applied to human therapeutics.

The organoid system circumvents many of these limitations. The potential of organoids to accurately capture the diversity of the endogenous cellular milieu comes from its ability to mimic the cytoarchitecture of the human brain by self-organizing into a multilayer aggregate where different cell types interact and signal to each other through close intercellular contacts (Figure 3). For example, ventricular radial glial cells communicate via N-cadherin and Notch signaling [80] and outer radial glia through LIFR-STAT3 [81] and mTOR [82] signaling, all of which profoundly influence their differentiation. Additionally, unlike the 2D system, organoids can be cultured for months at a time and even up to a year, maturing into electrophysiologically functional and biologically robust neural aggregates. This improved differentiation and maturation allow the development of long-range projections after organoids transplantation in mice, overcoming an obstacle that was of great consideration for the aforementioned medium spiny neuron development [83,84]. Indeed, Mansour et al. were able to show subpallial innervations following cortical transplantation of organoids in mice, with axonal projections to not only the striatum but to rostral regions as well. Additionally, projections spanned both hemispheres, with evidence of projections crossing the corpus callosum [83]. These works provide support and promise for the future of the organoid system. The remainder of this review will focus on the use and potential of SHH in the organoid system towards development and disease modeling.

## 6. The Future of In Vitro Technology

Scientists have continued to push the boundaries of empirical possibility, discovering strategies that more realistically render neural networks and neuronal circuitry. Beyond any previous in vitro system, organoids have better reproduced the range of cell phenotypes that reside in a specific brain region, and offer an assay to model the developmental interactions of two or more brain regions, recapitulating dynamic and complex migration patterns, regional crosstalk and brain circuitry. These single-region organoids or multi-region assembloids present remarkable new opportunities to examine circuit dysregulation and inhibitory/excitatory imbalances plaguing many neurodevelopmental disorders.

The organoid system reproduces the embryonic-fetal window of development, a unique and essential time span that is difficult to thoroughly investigate in many other model systems [12,85]. Researchers have used these multilayered diverse cellular aggregates to grow various brain regions and model many diseases, from autism spectrum disorder [86] to Rett syndrome [87] and Timothy syndrome [88]. Individually cultured dorsal and ventral forebrain regional organoids were fused to achieve whole forebrain-like organoids, complete with migrating DLX2+ inhibitory populations in the ventral-dorsal direction, mimicking in vivo neuronal behavior [88,89]; these experiments required the use of SHH agonists to stimulate development of the ventral forebrain and its migrating progenitors. With a recent influx of methods to foster ventral telencephalic-like organoids, the potential to study neurodevelopmental disorders affecting the basal ganglia, such as OCD and Tourette Syndrome, is perceivably high.

The union of independently generated regional organoids into assembloids achieves active cross-regional communication via processes such as cell migration; however, the cohesive nature of whole forebrain development and fluid transition of gene expression and cellular diversity is lost in assembloid strategies. As described earlier in this review, multi-axis neural patterning—dorso-ventral, medio-lateral, and anterio-posterior—is instructed by the SHH, WNT and BMP gradients endogenously established during neurogenesis (Figure 1B). The gradient fosters a precise topographic map that specifies regional identity within the human brain. Cederquist et al. recognized that current organoid technologies lack such holistic topography, although it is an essential feature of neurodevelopment. To mimic in vivo spatial organization of the human forebrain and achieve distinct positional domains, Cederquist et al. developed an inducible SHH-expressing hPSC line (iSHH) to initiate an inducible (rtTA-dependent) SHH gradient at a single pole of an organoid [90]. To create what Cederquist and colleagues referred to as an SHH organizer, 1000 iSHH cells were cultured independently to become a mini cellular aggregate, after which time a larger number of wild-type cells were added to assemble atop and fuse to the iSHH aggregate. The product created what the group referred to as a “chimeric spheroid”, with a much smaller ratio of iSHH cells to wild-type cells [90]. The organoid’s response to the resultant SHH protein product then becomes a function of distance: the further away cells are from the SHH organizer, the weaker the SHH-induced gene expression, thereby creating a gradient effect within the organoid and constructing a neuraxis.

The SHH organizer successfully produced four positional identities—dorsal, ventral, anterior and posterior [90]. Immunochemistry revealed the strong ventral expression of NKX2.1 in cells closest to the organizer, while those furthest away expressed the dorsal PAX6. Additionally, cells furthest from the organizer also expressed the anterior FOXG1, while those closest expressed the posterior OTX2. The distribution of SHH as its expression gradient traversed the organoid achieved the major subdivisions of the human forebrain, including the neocortex and ganglionic eminences MGE and LGE, and even anterior hypothalamic development [90].

The inducible system generated a single organoid representative of topographically ordered multiple cellular fates and regional identities. When comparing the 3D strategies currently available for neurodevelopment, the iSHH system has its advantages, as it more thoroughly captures the interconnectedness of the human forebrain. There are surely synchronous as well as sequential elements of neuro-pathway activation, the compound impact of which is essential for mimicking the remarkable complexities of neurodevelopment, and are likely missing from multi-organoid assembloid generation. A gradient-based system, as achieved with the SHH organizer, inches closer to in vivo conditions and is more capable of mimicking the dynamic cellular continuum achieved by the human forebrain. This is because morphogen’s concentrations inherently have differential ability to transcriptionally control gene expression, where precise concentrations either induce or silence gene expression and related cascades of signaling events, and hence the complexity that gradients can establish. Furthermore, gradient-based organoid systems can be combined with multi-regional assembloids, where the benefits of each are compounded, and may help to construct the most appropriate systems to use to answer the empirical question in the end.

The in vitro organoid system provides, at least in principle, a source for the investigation of a variety of developmental periods, with the potential to model therapeutic approaches, including modulation of gene expression, gene editing, and drug design. The tridimensional design offers a strategy that can enhance the empirical study of Parkinson’s and Huntington’s diseases mentioned previously, as well as a wide range of other neuropsychiatric disorders. Using either the assembloid or gradient approach, the development of dopaminergic projections from the human substantia nigra to the striatum can be investigated and their abnormalities characterized in a Parkinsonian organoid model. Long-range projections of medium spiny neurons from the striatum to the other regions of the basal ganglia can be modeled and their functionality tested in a Huntington’s organoid model. Tourette syndrome and OCD can be modeled by observing the development of the basal ganglia, and cortico-basal ganglia circuitry can be tracked in these diseases as well. These are only some of the possible applications of this system, but there are many exciting doors to be opened with this technology to advance our understanding and treatment of human disorders—we are only at the beginning.

## Figures and Tables

**Figure 1 cells-10-00914-f001:**
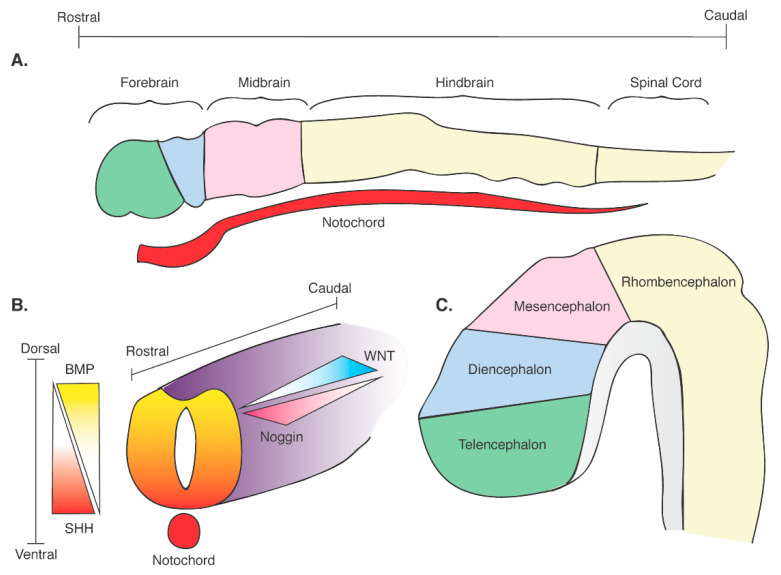
Schematic of neural tube formation and cerebral regional identity. (**A**) Regional definition of the neural plate with rostrocaudal axes and its contact with the SHH-releasing notochord. (**B**) Formation of the neural tube and its contact with the SHH-releasing notochord. Dorsoventral and rostrocaudal axes are defined with corresponding critical morphogenetic gradients: dorsal BMP; ventral SHH; rostral noggin; caudal WNT). (**C**) Development of the embryonic brain and divisions of the central nervous system. Modified from Marysia Placzek and James Briscoe, 2005 (ref. [54]).

**Figure 2 cells-10-00914-f002:**
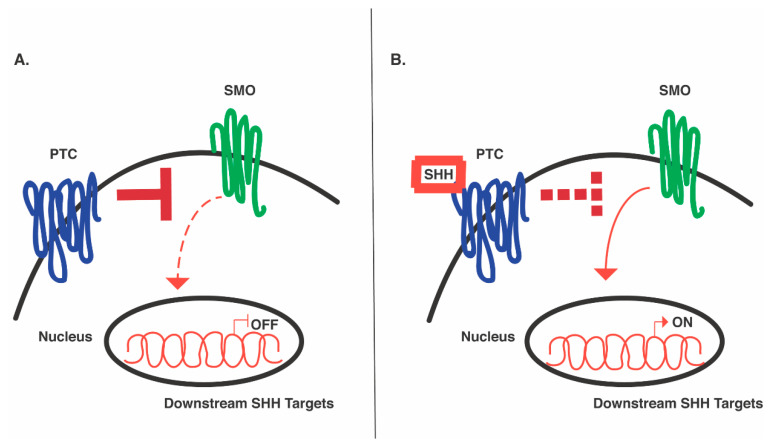
SHH signaling pathway. (**A**) SHH pathway in the absence of SHH ligand. PTC inhibits SMO activity, thereby inhibiting its ability to positively influence transcription (dashed lines). Nuclear transcription of downstream SHH targets is stopped. (**B**) SHH pathway in the presence of SHH ligand. SHH binds PTC, lifting its inhibition on SMO (dashed lines), thereby allowing SMO to positively influence transcription. Nuclear transcription of downstream targets is activated.

**Figure 3 cells-10-00914-f003:**
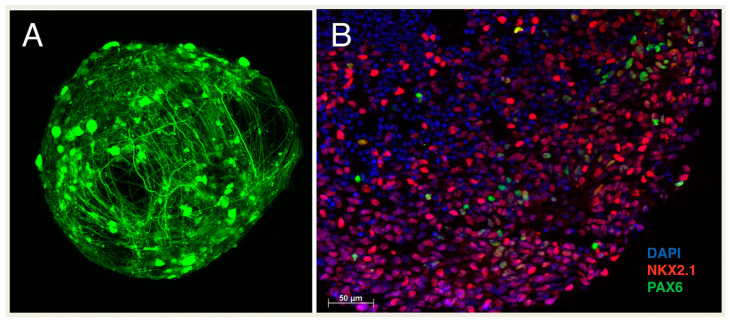
Human iPSC-derived organoid. (**A**): An iPSC-derived ventral telencephalic organoid, was transduced at 7 months in vitro with SYN-GFP to reveal post-mitotic neurons during development. Imaging was obtained by two-photon microscopy to show longevity and network complexity within these 3D aggregates. (**B**): Basal ganglia organoid imaged at terminal differentiation day 14, showing widespread protein expression of NKX2.1, a transcription factor expressed by ventral telencephalic progenitor cells, and minimal expression of PAX6, a transcription factor expressed by dorsal telencephalic progenitor cells.

## Data Availability

Not applicable.

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
