# Peer review of "Role of SHH in Patterning Human Pluripotent Cells towards Ventral Forebrain Fates"

_cells, 2021, doi:10.3390/cells10040914_

Round 1

Reviewer 1 Report

In this paper, Brady et al. present a review of recent papers on the role of organoid culture of human PSCs in understanding human neurodevelopment and its disorders. They especially focus on the SHH signaling for the regionalization of human organoids. This paper is well written and suitable for publishing, but they should address the following points.

Major points

  1. BMPs and WNTs, morphogens important for regionalization that interact with SHH, should be further described. In particular, it should be noted that SHH expression is regulated not only by the induction from notochord-derived SHH but also through repression by BMPs and WNTs (Ohkubo, Neurosci., 2002; Backman, Dev. Biol., 2005; Eto, Nat. Comm., 2020).
  2. In the "The use of SHH in in vitro disease modeling" section, almost all the work uses 2D culture. However, there is little mention of 2D culture in the introduction section. The section starting with "While these studies show" should be moved to the introduction section. Also, since 2D culture has been mainly used for the disease model work so far, the advantages of 2D culture should be emphasized.
  3. From p13, there are few examples of human organoids applications, which is the main topic of this paper. The authors can introduce the application of human organoids in a broader sense, not only in the area of SHH, such as Trujillo, Science, 2021.

Minor points

4. The section "The use of SHH in in vitro disease modeling" should be subdivided into some topics for easier reading.

Author Response

Response to reviewer 1

Major points

  1. BMPs and WNTs, morphogens important for regionalization that interact with SHH, should be further described. In particular, it should be noted that SHH expression is regulated not only by the induction from notochord-derived SHH but also through repression by BMPs and WNTs (Ohkubo, Neurosci., 2002; Backman, Dev. Biol., 2005; Eto, Nat. Comm., 2020).

Response: Thank you, we included these references and emphasized the important role that both BMPs and WNTs play in SHH expression (Page 7).

  1. In the "The use of SHH in in vitro disease modeling" section, almost all the work uses 2D culture. However, there is little mention of 2D culture in the introduction section. The section starting with "While these studies show" should be moved to the introduction section. Also, since 2D culture has been mainly used for the disease model work so far, the advantages of 2D culture should be emphasized.

Response: We have moved the identified paragraph to the introduction as suggested and further detailed the advantages of 2D culture to better capture the pros and cons of the system (Page 4 & 5), thank you.

  1. From p13, there are few examples of human organoids applications, which is the main topic of this paper. The authors can introduce the application of human organoids in a broader sense, not only in the area of SHH, such as Trujillo, Science, 2021.

 Response: We integrated the recent work by Trujillo et al to showcase a wide range of possibilities for organoids beyond the use of SHH (Page 5), thank you.

Minor points

  1. The section "The use of SHH in in vitro disease modeling" should be subdivided into some topics for easier reading.

Response: We have added subdivisions into this section, thank you.

Reviewer 2 Report

The review from Brady and Vaccarino explores the application of Sonic Hedgehog SHH to various stem cell methodologies and in vitro systems, including organoids. The article is interesting and covers exhaustively the SHH topic.

Title: The article seems more focused on the SHH system and all the in vitro systems and human INs developed to study it, while the title makes the reader think of a greater focus on organoids. Maybe a rephrase of the title could improve the match with the article content.

Page 4-5- first paragraph: Maybe the authors could explain a bit further the development/use of organoids, in particular the use of scaffolds for increasing the complexity of the organoids.

Page 4- please add briefly some details about the limitations you mention for ref 17,18

Page 7- Please specify the FGF8 acronym

Page 12- “The 2D system cultivates homogeneous cellular populations, unlike in vivo physiology, and this biological simplicity significantly limits cellular interactions thereby disrupting the overall development of the cells…”. This explanation is a bit limited. More than homogenous, the 2d systems are too simple and overall different from the in vivo situation, in fact, here cells lack all environmental stimuli (es. physical and mechanical), leading for example to cell polarization, in addition to cell-cell contacts. 

Page 12- “coupled with excessive contact with extracellular matrix”- It seems not completely correct to me, because the ECM is a complex 3d system and, strictly, is not present in standard cell cultures. Can the authors explain this sentence and add eventually references? If not please cancel it.

Minor points: Page 3, text after ref4- check the sentence, there is a strange repetition (typo)

Author Response

Response to reviewer 2.

The review from Brady and Vaccarino explores the application of Sonic Hedgehog SHH to various stem cell methodologies and in vitro systems, including organoids. The article is interesting and covers exhaustively the SHH topic.

Title: The article seems more focused on the SHH system and all the in vitro systems and human INs developed to study it, while the title makes the reader think of a greater focus on organoids. Maybe a rephrase of the title could improve the match with the article content.

Response: We changed the title to “Role of SHH in patterning human pluripotent cells towards ventral forebrain fates”.

Page 4-5- first paragraph: Maybe the authors could explain a bit further the development/use of organoids, in particular the use of scaffolds for increasing the complexity of the organoids.

Response: We have now included within the text the use of scaffolds to assist in organoid generation and specifically their use towards enhancing preparation reproducibility (Page 5).

Page 4- please add briefly some details about the limitations you mention for ref 17,18

Response: We have expanded upon the limitations mentioned in these references, including the time-costly nature of organoid development and their variable cellular proportions (Page 5).

Page 7- Please specify the FGF8 acronym

Response: We have fully described this term, thank you.

Page 12- “The 2D system cultivates homogeneous cellular populations, unlike in vivo physiology, and this biological simplicity significantly limits cellular interactions thereby disrupting the overall development of the cells…”. This explanation is a bit limited. More than homogenous, the 2d systems are too simple and overall different from the in vivo situation, in fact, here cells lack all environmental stimuli (es. physical and mechanical), leading for example to cell polarization, in addition to cell-cell contacts. 

Response: We have elaborated upon these concepts as suggested by the reviewer’s comments.

Page 12- “coupled with excessive contact with extracellular matrix”- It seems not completely correct to me, because the ECM is a complex 3d system and, strictly, is not present in standard cell cultures. Can the authors explain this sentence and add eventually references? If not please cancel it.

Response: We refer, here, to laminin and collagen, two major components of basement membranes and laminae that separate major tissue compartments, but are generally not present when cells directly contact each other. Both laminin and collagen are commonly used as coating surfaces to favor cell adhesion to the plastic dish in 2D culture. In a recent study comparing directly 2D and 3D systems we found that laminin-mediated integrin signaling was upregulated in monolayers versus organoids, while the reverse was true for cell adhesion molecules such as cadherins; blocking excessive integrin signaling reversed excessive cell proliferation in 2D cultures (Scuderi et al, 2020, Cell-to-Cell Adhesion and Neurogenesis in Human Cortical Development: A Study Comparing 2D Monolayers with 3D Organoid Cultures. Stem Cell Reports 2021, 16, 264-280). We have added a sentence clarifying this concept as well as the reference.

Minor points: Page 3, text after ref4- check the sentence, there is a strange repetition (typo)

Response: Thanks, we fixed it.